# Large Language Models Can Implement Policy Iteration

**Ethan Brooks[1], Logan Walls[2], Richard L. Lewis[2], Satinder Singh[1]**
[1]Computer Science and Engineering, University of Michigan
[2]Department of Psychology, University of Michigan
{ethanbro,logwalls,rickl,baveja}@umich.edu

## Abstract

In this work, we demonstrate a method for implementing policy iteration using a large language model. While the application of foundation models to Reinforcement Learning (RL) has received considerable attention, most approaches rely on either (1) the curation of expert demonstrations (through manual design or task-specific pretraining) or (2) adaptation to the task of interest using gradient methods (fine-tuning or training of adapter layers). Both of these techniques have drawbacks. Collecting demonstrations is labor-intensive, and algorithms that rely on them do not outperform the experts from which the demonstrations were derived. Gradient techniques are inherently slow, sacrificing the "few-shot" quality that makes in-context learning attractive to begin with. Our method demonstrates that a large language model can be used to implement policy iteration using the machinery of in-context learning, enabling it to learn to perform RL tasks without expert demonstrations or gradients. Our approach iteratively updates the contents of the prompt from which it derives its policy through trial-and-error interaction with an RL environment. In order to eliminate the role of in-weights learning (on which approaches like Decision Transformer (L. Chen et al. 2021) rely heavily), we demonstrate our method using Codex (M. Chen et al. 2021b), a language model that was not trained to perform the tasks which we are evaluating.

## 1 Introduction

In many settings, models implemented using a transformer or recurrent architecture will improve their performance as information accumulates in their context or memory. We refer to this phenomenon as "in-context learning." Brown et al. 2020b demonstrated a technique for inducing form of learning by prompting a large language model with a small number of input/output exemplars. An interesting property of in-context learning in the case of large pre-trained models (or "foundation models") is that the models are not directly trained to optimize a meta-learning objective, but demonstrate an emergent capacity to generalize (or at least specialize) to diverse downstream task-distributions (Wei et al. 2022b).

This remarkable capability has been studied in a variety of settings, including Reinforcement Learning (RL). However, most work in this area either (1) assumes access to expert demonstrations — collected either from human experts (Huang et al. 2022b; Baker et al. 2022), or domain-specific pre-trained RL agents (L. Chen et al. 2021; Lee et al. 2022; Janner et al. 2021; Reed et al. 2022; Xu et al. 2022). — or (2) relies on gradient-based methods — e.g. fine-tuning of the foundation models parameters as a whole (Lee et al. 2022; Reed et al. 2022; Baker et al. 2022) or newly training an adapter layer or prefix vectors while keeping the original foundation models frozen (X. L. Li et al. 2021; A. K. Singh et al. 2022; Karimi Mahabadi et al. 2022).

37th Conference on Neural Information Processing Systems (NeurIPS 2023).

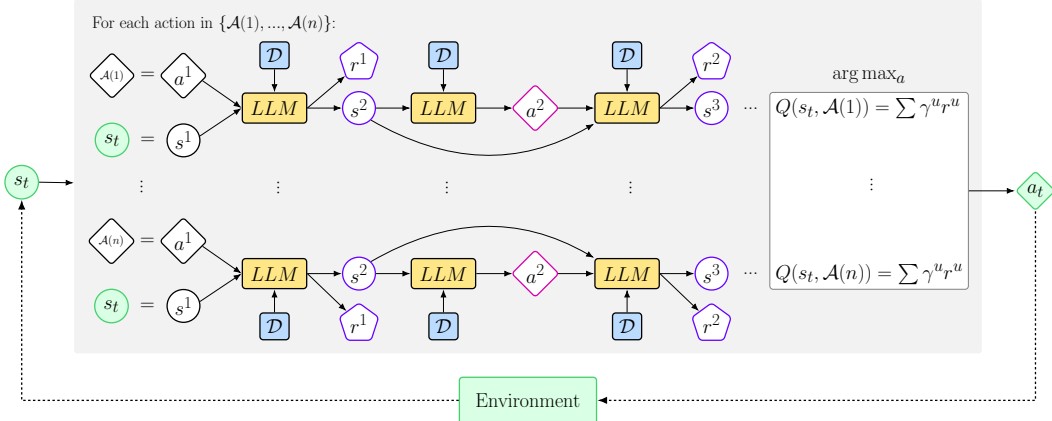

Figure 1: For each possible action $\mathcal{A}(1), \ldots, \mathcal{A}(n)$, the LLM generates a rollout by alternately predicting transitions and selecting actions. Q-value estimates are discounted sums of rewards. The action is chosen greedily with respect to Q-values. Both state/reward prediction and next action selection use trajectories from $\mathcal{D}$ to create prompts for the LLM. Changes to the content of $\mathcal{D}$ change the prompts that the LLM receives, allowing the model to improve its behavior over time.

Our work demonstrates an approach to in-context learning which relaxes these assumptions. Our method, In-Context Policy Iteration (ICPI), implements policy iteration using the prompt content, instead of the model parameters, as the locus of learning, thereby avoiding gradient methods. Furthermore, the use of policy iteration frees us from expert demonstrations because suboptimal prompts can be improved over the course of training.

We illustrate the method empirically on six small illustrative RL tasks— *chain, distractor-chain, maze, mini-catch, mini-invaders*, and *point-mass*—in which the method very quickly finds good policies. We also compare five pretrained Large Language Models (LLMs), including two different size models trained on natural language—OPT-30B and GPT-J—and three different sizes of a model trained on program code—two sizes of Codex as well as InCoder. On our six domains, we find that only the largest model (the `code-davinci-001` variant of Codex) consistently demonstrates learning.

## 2 Related Work

A common application of foundation models to RL involves tasks that have language input, for example natural language instructions/goals (D. Garg et al. 2022; Hill et al. 2020) or text-based games (Peng et al. 2021; I. Singh et al. 2021; Majumdar et al. 2020; Ammanabrolu et al. 2021). Another approach encodes RL trajectories into token sequences, and processes them with a foundation model, model representations as input to deep RL architectures (S. Li et al. 2022; Tarasov et al. 2022; Tam et al. 2022). Finally, a recent set of approaches (which we will focus on in this Related Work section) treat RL as a sequence modeling problem and use the foundation models itself to predict states or actions. In this related work section, we will focus a third set of recent approaches that treat reinforcement learning (RL) as a sequence modeling problem and utilize foundation models for state prediction, action selection, and task completion. We will organize our survey of these approaches based on how they elicit these RL-relevant outputs from the foundation models. In this respect the approaches fall under three broad categories: learning from demonstrations, specialization (via training or finetuning), and context manipulation (in-context learning).

### 2.1 Learning from demonstrations

Many recent sequence-based approaches to reinforcement learning use demonstrations that come either from human experts or pretrained RL agents. For example, Huang et al. 2022b use a frozen LLM as a planner for everyday household tasks by constructing a prefix from human-generated task instructions, and then using the LLM to generate instructions for new tasks. This work is extended by Huang et al. 2022a. Similarly, Ahn et al. 2022 use a value function that is trained on

human demonstrations to rank candidate actions produced by an LLM. Baker et al. 2022 use human demonstrations to train the foundation model itself: they use video recordings of human Minecraft players to train a foundation models that plays Minecraft. Works that rely on pretrained RL agents include Janner et al. 2021 who train a "Trajectory Transformer" to predict trajectory sequences in continuous control tasks by using trajectories generated by pretrained agents, and L. Chen et al. 2021, who use a dataset of offline trajectories to train a "Decision Transformer" that predicts actions from state-action-reward sequences in RL environments like Atari. Two approaches build on this method to improve generalization: Lee et al. 2022 use trajectories generated by a DQN agent to train a single Decision Transformer that can play many Atari games, and Xu et al. 2022 use a combination of human and artificial trajectories to train a Decision Transformer that achieves few-shot generalization on continuous control tasks. Reed et al. 2022 take task-generality a step farther and use datasets generated by pretrained agents to train a multi-modal agent that performs a wide array of RL (e.g. Atari, continuous control) and non-RL (e.g. image captioning, chat) tasks.

Some of the above works include non-expert demonstrations as well. L. Chen et al. 2021 include experiments with trajectories generated by random (as opposed to expert) policies. Lee et al. 2022 and Xu et al. 2022 also use datasets that include trajectories generated by partially trained agents in addition to fully trained agents. Like these works, our proposed method (ICPI) does not rely on expert demonstrations—but we note two key differences between our approach and existing approaches. Firstly, ICPI only consumes self-generated trajectories, so it does not require any demonstrations (like L. Chen et al. 2021 with random trajectories, but unlike Lee et al. 2022, Xu et al. 2022, and the other approaches reviewed above). Secondly, ICPI relies primarily on in-context learning rather than in-weights learning to achieve generalization (like Xu et al. 2022, but unlike L. Chen et al. 2021 & Lee et al. 2022). For discussion about in-weights vs. in-context learning see Chan et al. 2022.

## 2.2   Gradient-based training & finetuning on RL tasks

Many approaches that use foundation models for RL involve specifically training or fine-tuning on RL tasks. For example, Janner et al. 2021; L. Chen et al. 2021; Lee et al. 2022; Xu et al. 2022; Baker et al. 2022; Reed et al. 2022 all use models that are trained from scratch on tasks of interest, and A. K. Singh et al. 2022; Ahn et al. 2022; Huang et al. 2022a combine frozen foundation models with trainable components or adapters. In contrast, Huang et al. 2022b use frozen foundation models for planning, without training or fine-tuning on RL tasks. Like Huang et al. 2022b, ICPI does not update the parameters of the foundation model, but relies on the frozen model's in-context learning abilities. However, ICPI gradually builds and improves the prompts within the space defined by the given fixed text-format for observations, actions, and rewards (in contrast to Huang et al. 2022b, which uses the frozen model to select good prompts from a given fixed library of goal/plan descriptions).

## 2.3   In-Context learning

Several recent papers have specifically studied in-context learning. Laskin et al. 2022 demonstrates an approach to performing in-context reinforcement learning by training a model on complete RL learning histories, demonstrating that the model actually distills the improvement operator of the source algorithm. Chan et al. 2022 and S. Garg et al. 2022 provide analyses of the properties that drive in-context learning, the first in the context of image classification, the second in the context of regression onto a continuous function. These papers identify various properties, including "burstiness," model-size, and model-architecture, that in-context learning depends on. Y. Chen et al. 2022 studies the sensitivity of in-context learning to small perturbations of the context. They propose a novel method that uses sensitivity as a proxy for model certainty. Some recent work has explored iterative forms of in-context learning, similar to our own. For example, Shinn et al. 2023 and Madaan et al. 2023 use iterative self-refinement to improve the outputs of a large language model in a natural language context. These approaches rely on the ability of the model to examine and critique its own outputs, rather than using policy iteration as our method does.

## 3   Method

How can standard policy iteration make use of in-context learning? Policy iteration is either *model-based*—using a world-model to plan future trajectories in the environment—or *model-free*—inferring value-estimates without explicit planning. Both methods can be realized with in-context learning.

**Algorithm 2** Computing Q-values

---

1: **function** $Q(o_t, a, \mathcal{D})$
2:     $u \leftarrow t$
3:     $o^{(1)} = o_t$
4:     $a^{(1)} = a$
5:     **repeat**                                    ▷ All samples come from the experience buffer $\mathcal{D}$
6:         $\mathcal{D}_b \sim$ time-steps with action $a^{(u)}$                ▷ balancing terminal and non-terminal
7:         $b^{(u)} \sim \text{LLM}\left(\mathcal{D}_b, o^{(u)}, a^{(u)}\right)$
8:         $\mathcal{D}_r \sim$ time-steps with action $a^{(u)}$ and termination $b^{(u)}$            ▷ balancing reward
9:         $r^{(u)} \sim \text{LLM}\left(\mathcal{D}_r, o^{(u)}, a^{(u)}\right)$
10:         $\mathcal{D}_o \sim$ time-steps with action $a^{(u)}$ and termination $b^{(u)}$            ▷ no balancing
11:         $o^{(u+1)} \sim \text{LLM}\left(\mathcal{D}_o, o^{(u)}, a^{(u)}\right)$
12:         $\mathcal{D}_a \sim c$ recent trajectories
13:         $a^{(u+1)} \sim \text{LLM}\left(o^{(u+1)}, \mathcal{D}_a\right)$
14:         $u \leftarrow u + 1$
15:     **until** $b^{(u)}$ is terminal
16:     **return** $\sum_{k=1}^{u} \gamma^{k-1} r^{(k)}$
17: **end function**

---

We choose model-based learning because planned trajectories make the underlying logic of value-estimates explicit to our foundation model backbone, providing a concrete instantiation of a trajectory that realizes the values. This ties into recent work (Wei et al. 2022a; Nye et al. 2021) demonstrating that "chains of thought" can significantly improve few-shot performance of foundation models.

Model-based RL requires two ingredients, a rollout-policy used to act during planning and a world-model used to predict future rewards, terminations, and states. Since our approach avoids any mutation of the foundation model's parameters (this would require gradients), we must instead induce the rollout-policy and the world-model using in-context learning, i.e. by selecting appropriate prompts. We induce the rollout-policy by prompting the foundation model with trajectories drawn from the current (or recent) behavior policy (distinct from the rollout-policy). Similarly, we induce the world-model by prompting the foundation models with transitions drawn from the agent's history of experience. Note that our approach assumes access to some translation between the state-space of the environment and the medium (language, images, etc.) of the foundation models. This explains how an algorithm might plan and estimate values using a foundation model. It also explains how the rollout-policy approximately tracks the behavior policy.

How does the policy improve? When acting in the environment (as opposed to planning), we choose the action that maximizes the estimated Q-value from the current state (see Training Loop pseudocode, line 6). At time step $t$, the agent observes the state of the environment (denoted $s_t$) and executes action $a_t = \arg\max_{a \in \mathcal{A}} Q^{(\pi_t)}(o_t, a)$, where $\mathcal{A} = \{\mathcal{A}(1), \cdots, \mathcal{A}(n)\}$ denotes the set of $n$ actions

---

**Algorithm 1** Training Loop

---

1: **function** TRAIN(environment)
2:     initialize $\mathcal{D}$                        ▷ replay buffer containing full history of behavior
3:     **while** training **do**
4:         $o_0 \leftarrow$ Reset environment.
5:         **while** episode is not done **do**
6:             $a_t \leftarrow \arg\max_a Q(o_t, a, \mathcal{D})$                        ▷ policy improvement
7:             $o_{t+1}, r_t, b_t \leftarrow$ Execute $a_t$ in environment.
8:             $t \leftarrow t + 1$
9:         **end while**
10:         $\mathcal{D} \leftarrow \mathcal{D} \cup (o_0, a_0, r_0, b_0, o_1, \ldots, o_t, a_t, r_t, b_t, o_{t+1})$        ▷ add trajectory to buffer
11:     **end while**
12: **end function**

---

available, $\pi_t$ denotes the policy of the agent at time step $t$, and $Q^{(\pi)}$ denotes the Q-estimate for policy $\pi$. Taking the greedy ($\arg\max$) actions with respect to $Q^{\pi_t}$ implements a new and improved policy.

**Computing Q-values** This section provides details on the prompts that we use in our computation of Q-values (see Computing Q-values pseudocode & Figure 1). During training, we maintain a buffer $\mathcal{D}$ of transitions experienced by the agent. To compute $Q^{(\pi_t)}(o_t, a)$ at time step $t$ in the real-world we rollout a simulated trajectory $o^{(1)} = o_t, a^{(1)} = a, r^{(1)}, o^{(2)}, a^{(2)}, r^{(2)}, \cdots, o^{(T)}, a^{(T)}, r^{(T)}, o^{(T+1)}$ by predicting, at each simulation time step $u$: reward $r^{(u)} \sim \mathrm{LLM}\left(\mathcal{D}_r, o^{(u)}, a^{(u)}\right)$; termination $b^{(u)} \sim \mathrm{LLM}\left(\mathcal{D}_b, o^{(u)}, a^{(u)}\right)$; observation $o^{(u+1)} \sim \mathrm{LLM}\left(\mathcal{D}_o, o^{(u)}, a^{(u)}\right)$; action $a^{(1)} \sim \mathrm{LLM}\left(\mathcal{D}_a, o^{(u)}\right)$. Termination $b^{(u)}$ decides whether the simulated trajectory ends at step $u$.

The prompts $\mathcal{D}_r, \mathcal{D}_b$ contain data sampled from the replay buffer. For each prompt, we choose some subset of replay buffer transitions, shuffle them, convert them to text (examples are provided in table 4.1) and clip the prompt at the 4000-token Codex context limit. We use the same method for $\mathcal{D}_a$, except that we use random trajectory subsequences.

In order to maximize the relevance of the prompt contents to the current inference we select transitions using the following criteria. $\mathcal{D}_b$ contains $(o_k, a_k, b_k)$ tuples such that $a_k$ equals $a^{(u)}$, the action for which the LLM must infer termination. $\mathcal{D}_r$ contains $(o_k, a_k, r_k)$ tuples, again constraining $a_k = a^{(u)}$ but also constraining $b_k = b^{(k)}$ — that the tuple corresponds to a terminal time-step if the LLM inferred $b^{(u)} = $ true, and to a non-terminal time-step if $b^{(u)} = $ false. For $\mathcal{D}_o$, the prompt includes $(o_k, a_k o_{k+1})$ tuples with $a_k = a^{(u)}$ and $b_k = $ false (only non-terminal states need to be modelled).

We also maintain a balance of certain kinds of transitions in the prompt. For termination prediction, we balance terminal and non-terminal time-steps. Since non-terminal time-steps far outnumber terminal time-steps, this eliminates a situation wherein the randomly sampled prompt time-steps are entirely non-terminal, all but ensuring that the LLM will predict non-termination. Similarly, for reward prediction, we balance the number of time-steps corresponding to each reward value stored in $\mathcal{D}$. In order to balance two collections of unequal size, we take the smaller and duplicate randomly chosen members until the sizes are equal.

In contrast to the other predictions, we condition the rollout policy on trajectory subsequences, not individual time-steps. Prompting with sequences better enables the foundation model to apprehend the logic behind a policy. Trajectory subsequences consist of $(o_k, a_k)$ pairs, randomly clipped from the $c$ most recent trajectories. More recent trajectories will, in general demonstrate higher performance, since they come from policies that have benefited from more rounds of improvement.

In contrast to the other predictions, we condition the rollout policy on trajectory subsequences, not individual time-steps. Prompting with sequences better enables the foundation model to apprehend the logic behind a policy. Trajectory subsequences consist of $(o_k, a_k)$ pairs, randomly clipped from the $c$ most recent trajectories. More recent trajectories will, in general demonstrate higher performance, since they come from policies that have benefited from more rounds of improvement.

Finally, the Q-value estimate is simply the discounted sum of rewards for the simulated episode. Given this description of Q-value estimation, we now return to the concept of policy improvement.

**Policy-Improvement** The $\arg\max$ (line 6 of Algorithm 1) drives policy improvement in ICPI. Critically this is not simply a one-step improvement but a mechanism that builds improvement on top of improvement. This occurs through a cycle in which the $\arg\max$ improves behavior. The improved behavior is stored in the buffer $\mathcal{D}$, and then used to condition the rollout policy. This improves the returns generated by the LLM during planning rollouts. These improved rollouts improve the Q-estimates for each action. Completing the cycle, this improves the actions chosen by the $\arg\max$. Because this process feeds into itself, it can drive improvement without bound until optimality is achieved.

Note that this process takes advantage of properties specific to in-context learning. In particular, it relies on the assumption that the rollout policy, when prompted with trajectories drawn from a mixture of policies, will approximate something like an average of these policies. Given this assumption, the rollout policy will improve with the improvement of the mixture of policies from which its prompt-trajectories are drawn. This results in a kind of rapid policy improvement that works without any use of gradients.

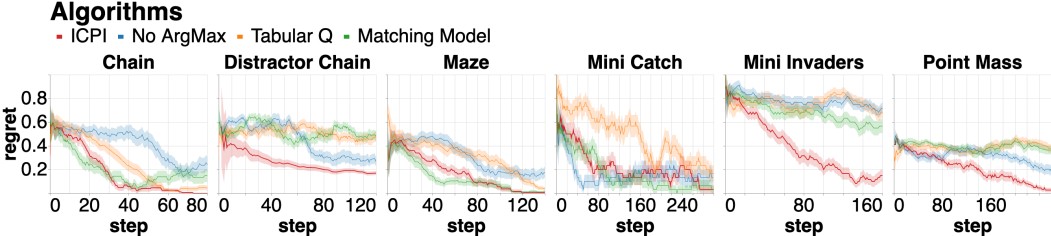

Figure 2: Comparison of ICPI with three baselines, "No $\arg\max$", "Tabular Q," and "Nearest Neighbor." The $y$-axis depicts regret (normalized between 0 and 1), computed relative to an optimal return with a discount-factor of 0.8. The $x$-axis depicts time-steps during training. Error bars are standard errors from four seeds.

**Prompt-Format**  The LLM cannot take non-linguistic prompts, so our algorithm assumes access to a textual representation of the environment—of states, actions, terminations, and rewards—and some way to recover the original action, termination, and reward values from their textual representation (we do not attempt to recover states). Since our primary results use the Codex language model (see Table 1), we use Python code to represent these values (examples are available in Table 2 in the appendix).

In our experiments, we discovered that the LLM world-model was unable to reliably predict rewards, terminations, and next-states on some of the more difficult environments. We experimented with providing domain *hints* in the form of prompt formats that make explicit useful information — similar to Chain of Thought Prompting (Wei et al. 2022a). For example, for the *chain* domain, the hint includes an explicit comparison (`==` or `!=`) of the current state with the goal state. Note that while hints are provided in the initial context, the LLM must infer the hint content in rollouts generated from this context.

We use a consistent idiom for rewards and terminations, namely `assert reward == x` and `assert done` or `assert not done`. Some decisions had to be made when representing states and actions. In general, we strove to use simple, idiomatic, concise Python. On the more challenging environments, we did search over several options for the choice of hint. For examples, see Table 2. We anticipate that in the future, stronger foundation models will be increasingly robust to these decisions.

## 4 Experiments

We have three main goals in our experiments: (1) Demonstrate that the agent algorithm can in fact quickly learn good policies, using pretrained LLMs, in a set of six simple illustrative domains of increasing challenge; (2) provide evidence through an ablation that the policy-improvement step—taking the $\arg\max$ over Q-values computed through LLM rollouts—accelerates learning; and (3) investigate the impact of using different LLMs (see Table 1)—different sizes and trained on different data, in particular, trained on (mostly) natural language (GPT-3 and GPT-J) vs. program code (Codex and InCoder). We next describe the six domains and their associated prompt formats, and then describe the experimental methodology and results.

### 4.1  Domains and prompt format

**Chain.**  In this environment, the agent occupies an 8-state chain. The agent has three actions: *Left*, *right*, and *try goal*. The *try goal* action always terminates the episode, conferring a reward of 1 on state 4 (the goal state) and 0 on all other states. Episodes also terminate after 8 time-steps. States are represented as numbers from 0 to 7, as in `assert state == n`, with the appropriate integer substituted for `n`. The actions are represented as functions `left()`, `right()`, and `try_goal()`. For the hint, we simply indicate whether or not the current state matches the goal state, 4.

**Distractor Chain.**  This environment is an 8-state chain, identical to the *chain* environment, except that the observation is a *pair* of integers, the first indicating the true state of the agent and the second acting as a distractor which transitions randomly within $\{0, \ldots, 7\}$. The agent must therefore learn to ignore the distractor integer and base its inferrences on the information contained in the first integer.

Aside from the addition of this distractor integer to the observation, all text representations and hints are identical to the *chain* environment.

**Maze.** The agent navigates a small $3 \times 3$ gridworld with obstacles. The agent can move *up*, *down*, *left*, or *right*. The episode terminates with a reward of 1 once the agent navigates to the goal grid, or with a reward of 0 after 8 time-steps. This environment tests our algorithms capacity to handle 2-dimensional movement and obstacles, as well as a 4-action state-space. We represent the states as namedtuples — C(x, y), with integers substituted for x and y. Similar to *chain*, the hint indicates whether or not the state corresponds to the goal state.

**Mini Catch.** The agent operates a paddle to catch a falling ball. The ball falls from a height of 5 units, descending one unit per time step. The paddle can *stay* in place (not move), or move *left* or *right* along the bottom of the 4-unit wide plane. The agent receives a reward of 1 for catching the ball and 0 for other time-steps. The episode ends when the ball's height reaches the paddle regardless of whether or not the paddle catches the ball. We chose this environment specifically to challenge the action-inference/rollout-policy component of our algorithm. Specifically, note that the success condition in Mini Catch allows the paddle to meander before moving under the ball—as long as it gets there on the final time-step. Successful trajectories that include movement *away* from the ball thus make a good rollout policies more challenging to learn (i.e., elicit from the LLM via prompts). Again, we represent both the paddle and the ball as namedtuples C(x, y) and we represent actions as methods of the paddle object: paddle.stay(), paddle.left(), and paddle.right(). For the hint, we call out the location of the paddle's $x$-position, the ball's $x$-position, the relation between these positions (which is larger than which, or whether they are equal) and the ball's $y$-position. Table 2 in the appendix provides an example. We also include the text ball.descend() to account for the change in the ball's position between states.

**Mini Invaders.** The agent operates a ship that shoots down aliens which descend from the top of the screen. At the beginning of an episode, two aliens spawn at a random location in two of four columns. The episode terminates when an alien reaches the ground (resulting in 0 reward) or when the ship shoots down both aliens (the agent receives 1 reward per alien). The agent can move *left*, *right*, or *shoot*. This domain highlights ICPI's capacity to learn incrementally, rather than discovering an optimal policy through random exploration and then imitating that policy, which is how our "No $\arg\max$" baseline learns (see Comparison of ICPI with baseline algorithms). ICPI initially learns to shoot down one alien, and then builds on this good but suboptimal policy to discover the better policy of shooting down both aliens. In contrast, random exploration takes much longer to discover the optimal policy and the "No $\arg\max$" baseline has only experienced one or two successful trajectories by the end of training.

We represent the ship by its namedtuple coordinate (C(x, y)) and the aliens as a list of these namedtuples. When an alien is shot down, we substitute None for the tuple, as in aliens == [C(x, y), None]. We add the text for a in aliens:  a.descend() in order to account for the change in the alien's position between states.

**Point-Mass.** A point-mass spawns at a random position on a continuous line between $-6$ and $+6$ with a velocity of 0. The agent can either *accelerate* the point-mass (increase velocity by 1) or *decelerate* it (decrease the velocity by 1). The point-mass position changes by the amount of its velocity each timestep. The episode terminates with a reward of 1 once the point-mass is between $-2$ and $+2$ and its velocity is 0 once again. The episode also terminates after 8 time-steps. This domain tests the algorithm's ability to handle continuous states.

States are represented as assert pos == p and vel == v, substituting floats rounded to two decimals for p and v. The actions are accel(pos, vel) and decel(pos, vel). The hint indicates whether the success conditions are met, namely the relationship of pos to $-2$ and $+2$ and the whether or not vel == 0. The hint includes identification of the aliens' and the ship's $x$-positions as well as a comparison between them.

## 4.2 Experiment Methodology and Results

**Methodology and evaluation.** For the results, we record the agent's regret over the course of training relative to an optimal policy computed with a discount factor of 0.8. For all experiments

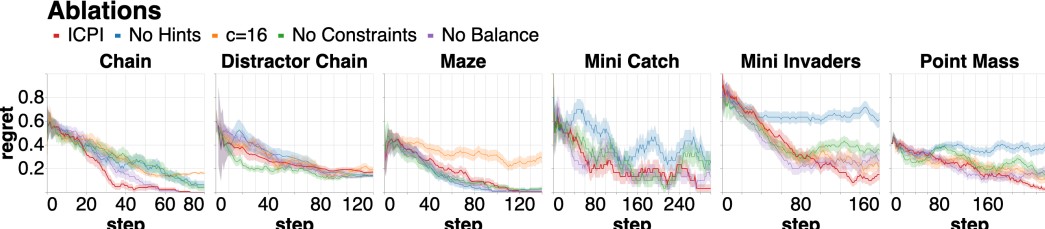

Figure 3: Comparison of ICPI with ablations. The $y$-axis depicts regret (normalized between 0 and 1), computed relative to an optimal return with a discount-factor of 0.8. The $x$-axis depicts time-steps during training. Error bars are standard errors from four seeds.

$c = 8$ (the number of most recent successful trajectories to include in the prompt). We did not have time for hyperparameter search and chose this number based on intuition. However the $c = 16$ baseline demonstrates results when this hyperparameter is doubled. All results use 4 seeds.

For both versions of Codex, we used the OpenAI Beta under the API Terms of Use. For GPT-J (B. Wang et al. 2021) , InCoder (Fried et al. 2022) and OPT-30B (Zhang et al. 2022), we used the open-source implementations from Huggingface Transformers (Wolf et al. 2020), each running on one Nvidia A40 GPU. All language models use a sampling temperature of 0.1. Code for our implementation is available at `https://github.com/ethanabrooks/icpi`.

**Comparison of ICPI with baseline algorithms.** We compare ICPI with three baselines (Fig. 2).

The "No $\arg\max$" baseline learns a good policy through random exploration and then imitates this policy. This baseline assumes access to a "success threshold" for each domain — an undiscounted cumulative return greater than which a trajectory is considered successful. The action selection mechanism emulates ICPI's rollout policy: prompting the LLM with a set of trajectories and eliciting an action as output. For this baseline, we only include trajectories in the prompt whose cumulative return exceeds the success threshold. Thus the policy improves as the number of successful trajectories in the prompt increases over time. Note that at the start of learning, the agent will have experienced too few successful trajectories to effectively populate the policy prompt. In order to facilitate exploration, we act randomly until the agent experiences 3 successes.

"Tabular Q" is a standard tabular Q-learning algorithm, which uses a learning rate of $1.0$ and optimistically initializes the Q-values to $1.0$.

"Matching Model" is a baseline which uses the trajectory history instead of an LLM to perform modelling. This baseline searches the trajectory buffer for the most recent instance of the current state, and in the case of transition/reward/termination prediction, the current action. If a match is found, the model outputs the historical value (e.g. the reward associated with the state-action pair found in the buffer). If no match is found, the modelling rollout is terminated. Recall that ICPI breaks ties randomly during action selection so this will often lead to random action selection.

As our results demonstrate, only ICPI learns good policies on all domains. We attribute this advantage to ICPI's ability to generalize from its context to unseen states and state/action pairs (unlike "Tabular Q" and "Matching Model"). Unlike "No $\arg\max$" ICPI is able to learn progressively, improving the policy before experiencing good trajectories.

**Ablation of ICPI components.** With these experiments, we ablate those components of the algorithm which are not, in principle, essential to learning (Fig. 3). "No Hints" ablates the hints described in the Prompt-Format paragraph. "No Balance" removes the balancing of different kinds of time-steps described in the Computing Q-values paragraph (for example, $\mathcal{D}_b$ is allowed to contain an unequal number of terminal and non-terminal time-steps). The "No Constraints" baseline removes the constraints on these time-steps described in the same paragraph. For example, $\mathcal{D}_r$ is allowed to contain a mixture of terminal and non-terminal time-steps (regardless of the model's termination prediction). Finally, "$c = 16$" prompts the rollout policy with the last 16 trajectories (instead of the last 8, as in ICPI). We find that while some ablations match ICPI's performance in several domains, none match its performance on all six.

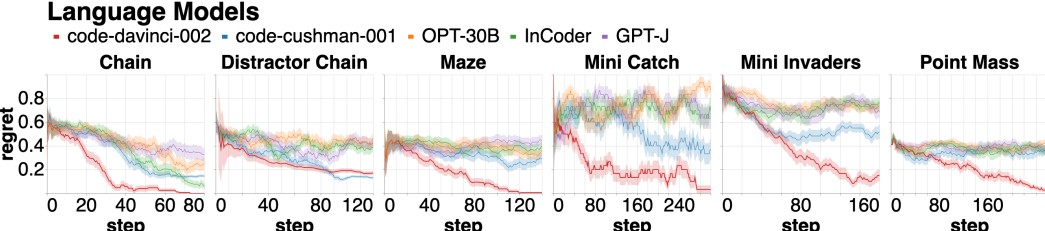

Figure 4: Comparison of different language models used to implement ICPI. The $y$-axis depicts regret (normalized between 0 and 1), computed relative to an optimal return with a discount-factor of 0.8. The $x$-axis depicts time-steps of training. Error bars are standard errors from four seeds.

**Comparison of Different Language Models.**    While our lab lacks the resources to do a full study of scaling properties, we did compare several language models of varying size (Fig. 4). See Table 1 for details about these models. Both `code-davinci-002` and `code-cushman-001` are variations of the Codex language model. The exact number of parameters in these models is proprietary according to OpenAI, but M. Chen et al. 2021a describes Codex as fine-tuned from GPT-3 (Brown et al. 2020a), which contains 185 billion parameters. As for the distinction between the variations, the OpenAI website describes `code-cushman-001` as "almost as capable as Davinci Codex, but slightly faster."

We found that the rate of learning and final performance of the smaller models fell significantly short of Codex on all but the simplest domein, *chain*. Examining the trajectories generated by agents trained using these models, we noted that in several cases, they seemed to struggle to apprehend the underlying "logic" of successful trajectories, which hampered the ability of the rollout policy to produce good actions. Since these smaller models were not trained on identical data, we are unable to isolate the role of size in these results. However, the failure of all of these smaller models to learn suggests that size has some role to play in performance. We conjecture that larger models developed in the future may demonstrate comparable improvements in performance over our Codex model.

**Limitations**    ICPI can theoretically work on any control task with discrete actions, due to the guarantees associated with policy iteration. However, since our implementation uses Codex, the domains in our paper were limited by the ability to encode states as text and to fit these encodings in the model's context window. Moreover, Codex demonstrated a limited ability to predict transitions and actions in more complex domains. As sequence models mature, we anticipate that more domains will become tractable for ICPI. We also note that reliance on the proprietary OpenAPI API limits exact reproduction of these results.

**Societal Impacts**    An extensive literature (Tamkin et al. 2021; Abid et al. 2021; Liang et al. 2021; Pan et al. 2023) has explored the possible positive and negative impacts of LLMs. Some of this work has explored mitigation strategies. In extending LLMs to RL, our work inherits these benefits and challenges. We highlight two concerns: the use of LLMs to spread misinformation and the detrimental carbon cost of training and using these models.

## 5   Conclusion

Our main contribution is a method for implementing policy iteration algorithm using Large Language Models and the mechanism of in-context learning. The algorithm uses a foundation models as both a world model and policy to compute Q-values via rollouts. Although we presented the method here as text-based, it is general enough to be applied to any foundation models that works through prompting, including multi-modal models like Reed et al. 2022 and Seo et al. 2022. In experiments we showed that the algorithm works in six illustrative domains imposing different challenges for ICPI, confirming the benefit of the LLM-rollout-based policy improvement. While the empirical results are preliminary, we believe the approach provides an important new way to use LLMs that will increase in effectiveness as the models themselves become more powerful.

Table 1: Pretrained Large Language Models (LLMs) Used in Experiments

| Model | Parameters | Training data |
|-------|-----------|---------------|
| GPT-J (B. Wang et al. 2021) | 6 billion | "The Pile" (Leo Gao et al. 2020), an 825GB English corpus incl. Wikipedia, GitHub, academic pubs |
| InCoder (Fried et al. 2022) | 6.7 billion | 159 GB of open-source StackOverflow code |
| OPT-30B (Zhang et al. 2022) | 30 billion | 180B tokens of predominantly English data including "The Pile" (Leo Gao et al. 2020) and "PushShift.io Reddit" (Baumgartner et al. 2020) |
| Codex (M. Chen et al. 2021a) | 185 billion | 179 GB of GitHub code |

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

# Appendix

## A  Example prompts for each domain

| | |
|---|---|
| **Chain** | ```python
assert state == 6 and state != 4
state = left()
assert reward == 0
assert not done
``` |
| **Distractor** | ```python
assert state == (6, 3) and state != (4, 3)
state = left()
assert reward == 0
assert not done
``` |
| **Maze** | ```python
assert state == C(i=2, j=1)
    and state != C(i=1, j=0)
state, reward = left()
assert reward == 0
assert not done
``` |
| **Mini Catch** | ```python
assert paddle == C(2, 0)
    and ball == C(0, 4)
    and paddle.x == 2 and ball.x == 0
    and paddle.x > ball.x
    and ball.y == 4
reward = paddle.left()
ball.descend()
assert reward == 0
assert not done
``` |
| **Mini Invaders** | ```python
assert ship == C(2, 0)
    and aliens == [C(3, 5), C(1, 5)]
    and (ship.x,
        aliens[0].x,
        aliens[1].x) == (2, 3, 1)
    and ship.x < aliens[0].x
    and ship.x > aliens[1].x
ship.left()
assert reward == 0
for a in aliens:
    a.descend()
assert not done
``` |
| **Point-Mass** | ```python
assert pos == -3.45 and vel == 0.00
    and pos < -2 and vel == 0|
pos, vel = decel(pos, vel)
assert reward == 0
assert not done
``` |

Table 2: This table provides example prompts for each domain, showcasing the text format and hints. Hints are highlighted in grey.

# B Tabular Q-Learning Hyperparameter Sweeps

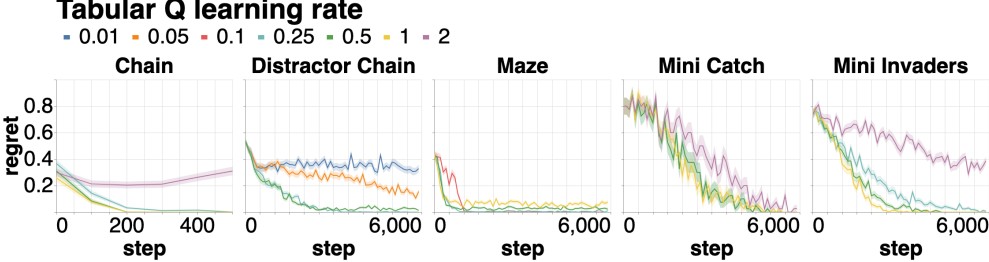

This figure presents learning curves using the standard Tabular Q-Learning algorithm. Given an appropriately chosen learning rate, this algorithm does converge, but this can take several thousand timesteps on all but the simplest domains.

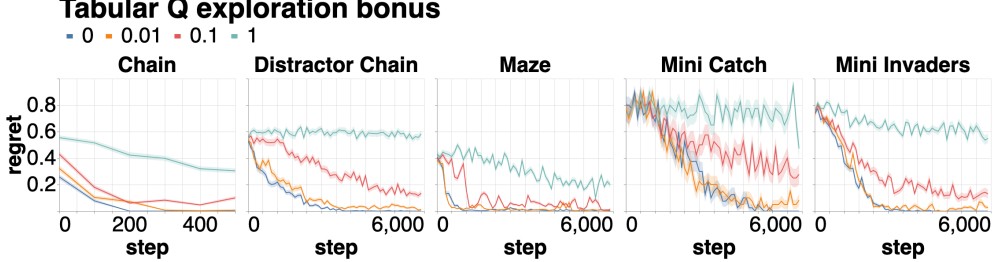

This figure presents learning curves for Tabular Q-Learning with a count-based exploration bonus. Actions are chosen using the following formulation: $a \leftarrow \arg\max_{a'} Q(s, a') + k/(1 + N(s, a'))$ where $k$ is the exploration bonus hyperparameter, assuming values $\{0, 0.01, 0.1, 1\}$ in the graph, and $N(s, a')$ is the visitation count for the state-action $(s, a')$.

# C Proximal Policy Optimization Hyperparameter Sweeps

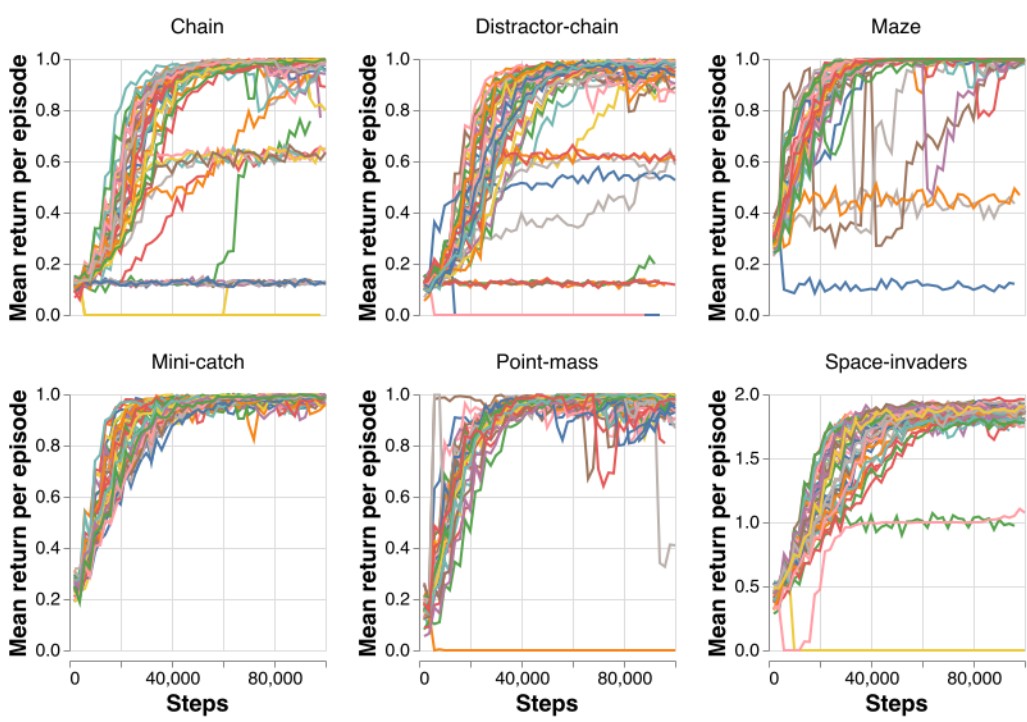

This figure presents learning curves for Proximal Policy Optimization (PPO) (Schulman et al. 2017), a deep RL algorithm trained using gradient descent. These runs search the following hyperparameters:

| | |
|---|---|
| **Number of Hidden Layers** | 1, 2 |
| **Hidden Size** | 256, 512, 1024 |
| **Actor Learning Rate** | 0.001, 0.002, 0.005 |
| **Critic Learning Rate** | 0.0001, 0.0005, 0.001 |

As the graphs indicate, PPO takes several thousand steps to converge, even on simple domains.

## D  Random Prediction Ablation

This ablation studies the contribution that each prediction type makes to ICPI's ability to learn. To this end, for prediction type — observation, termination, action, and reward — we ran an ablation that substitutes a random, valid prediction for the prediction made by the LLM. For example, for actions, we first sample a random action from the action space of the environment and then apply the standard string formatting used in other parts of the algorithm. As figure 5 indicates, on Chain, the simplest of our environments, none of the ablations are able to learn.

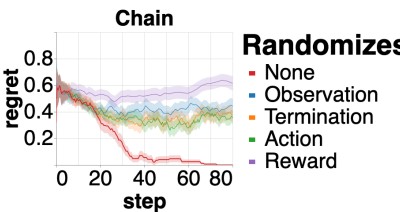

Figure 5: This figure presents learning curves for the random prediction ablation described above on the Chain domain. The Y-axis represents regret and the X-axis represents timesteps in the environment. Each colored line refers to a different prediction which was randomized. The red line is the original ICPI learning curve, included for reference.

## E  Error Analysis

In this section we present examples of errors made by ICPI on each domain. In each subsection, we will examine one rollout generated by the Codex LLM that causes the agent to choose suboptimal behavior in the actual environment.

### Chain

Here, we analyze an example in which the rollouts incorrectly estimate equal values for all three actions. Each line in the code snippet represents actual output from the LLM, excluding comments which are added for clarity. The agent's state is 5 (recall that the goal state is 4). For the `left()` action, the model generates the following rollout:

```
assert state == 5 and state != 4  # the current, starting state
state = left()  # a new state is the result of the left() action
assert not done
assert reward == 0
assert state == 4 and state == 4  # The new state is state 4, the goal state

# Here the rollout policy unexpectedly chooses to move right
# instead of trying the goal.
state = right()
assert not done
assert reward == 0
assert state == 5 and state != 4

# Again the rollout policy outputs a suboptimal action:
reward = try_goal(state)
assert done  # try_goal always terminates an episode with 0 reward
assert reward == 0
```

Note that the model outputs the action `right()` when the agent is in goal state 4, and then outputs action `try_goal()`. As a result the rollout estimates value 0 for an action which has value $\gamma$ under the optimal policy (one timestep away from a terminal state with reward 1).

The estimate for action `try_goal()` is accurate in state 5:

```
assert state == 5 and state != 4  # the current, starting state
reward = try_goal(state)
assert done
assert reward == 0
```

However the rollout also underestimates the value of action `right()`:

```
assert state == 5 and state != 4  # the current, starting state
state = right()
assert not done
assert reward == 0
assert state == 6 and state != 4
reward = try_goal(state)  # left() would be the optimal policy, not try_goal().
assert done
assert reward == 0
```

**Distracter Chain**

In this example the agent starts in state `(2, 3)`. Recall that in this environment, the second digit in the tuple is chosen randomly and does not affect environment transitions or reward.

In the first of the three rollouts, the first action is `left()`. Since the goal is state `(4, *)`, this action is already suboptimal. However, we see that the agent continues to behave sub-optimally throughout the rest of the rollout by continuously choosing action `left()` instead of the optimal action `right()`. This occurs because at this stage of training, the agent has mostly taken `left()` actions and as a result, the rollout policy prompt mostly contains `left()` actions.

```
assert state == [2, 3] and state != [4, 3]
state = left()
assert not done
assert reward == 0
assert state == [1, 2] and state != [4, 2]
state = left()  # The optimal action is right()
assert not done
assert reward == 0
assert state == [0, 0] and state != [4, 0]
state = left()  # The optimal action is right()
assert not done
assert reward == 0
assert state == [0, 0] and state != [4, 0]
state = left()  # The optimal action is right()
assert not done
assert reward == 0
assert state == [0, 0] and state != [4, 0]
state = left()  # The optimal action is right()
assert not done
assert reward == 0
assert state == [0, 0] and state != [4, 0]
state = left()  # The optimal action is right()
assert not done
assert reward == 0
assert state == [0, 1] and state != [4, 1]
state = left()  # completion
```

The rollout for the `try_goal()` action is correct:

```
assert state == [2, 3] and state != [4, 3]
reward = try_goal(state)
assert done
assert reward == 0  # completion
```

Here the initial action is optimal: `right()`. However, we once again see that the rollout policy is suboptimal because it favors the `left()` action over the optimal `right()` action.

```
assert state == [2, 3] and state != [4, 3]
state = right()
assert not done
assert reward == 0
assert state == [3, 7] and state != [4, 7]
```

```
state = left()
assert not done
assert reward == 0
assert state == [2, 4] and state != [4, 4]
state = left()
assert not done
assert reward == 0
assert state == [1, 7] and state != [4, 7]
state = left()
assert not done
assert reward == 0
assert state == [0, 0] and state != [4, 0]
state = left()
assert not done
assert reward == 0
assert state == [0, 0] and state != [4, 0]
state = left()
assert not done
assert reward == 0
assert state == [0, 0] and state != [4, 0]
state = left()
```

This causes the model to inaccurately produce a value estimate of 0 for the `right()` action.

**Maze**

Similar to the previous example, we study an example for maze early in training, when the rollout policy is still quite suboptimal. As a result, the rollout policy does not discover the reward, causing the rollouts to estimate equal value for each of the four actions.

The rollouts all start at state $(1, 1)$.

```
assert state == C(i=1, j=1) and state != C(i=1, j=0)
state, reward = left()
assert not done
assert reward == 0
assert state == C(i=1, j=0) and state != C(i=1, j=0)
state, reward = right()
assert not done
assert reward == 0
assert state == C(i=1, j=1) and state != C(i=1, j=0)
state, reward = up()
assert not done
assert reward == 0
assert state == C(i=0, j=1) and state != C(i=1, j=0)
state, reward = down()
assert not done
assert reward == 0
assert state == C(i=1, j=1) and state != C(i=1, j=0)
state, reward = up()
assert not done
assert reward == 0
assert state == C(i=0, j=1) and state != C(i=1, j=0)
state, reward = down()
assert not done
assert reward == 0
assert state == C(i=1, j=1) and state != C(i=1, j=0)
state, reward = up()

assert state == C(i=1, j=1) and state != C(i=1, j=0)
state, reward = down()
assert not done
assert reward == 0
assert state == C(i=2, j=1) and state != C(i=1, j=0)
state, reward = down()
assert not done
assert reward == 0
assert state == C(i=2, j=1) and state != C(i=1, j=0)
state, reward = left()
assert not done
assert reward == 0
assert state == C(i=2, j=0) and state != C(i=1, j=0)
state, reward = right()
assert not done
assert reward == 0
assert state == C(i=2, j=1) and state != C(i=1, j=0)
state, reward = right()
assert not done
```

```
assert reward == 0
assert state == C(i=2, j=2) and state != C(i=1, j=0)
state, reward = right()
assert not done
assert reward == 0
assert state == C(i=2, j=2) and state != C(i=1, j=0)
state, reward = right()

assert state == C(i=1, j=1) and state != C(i=1, j=0)
state, reward = up()
assert not done
assert reward == 0
assert state == C(i=0, j=1) and state != C(i=1, j=0)
state, reward = down()
assert not done
assert reward == 0
assert state == C(i=1, j=1) and state != C(i=1, j=0)
state, reward = up()
assert not done
assert reward == 0
assert state == C(i=0, j=1) and state != C(i=1, j=0)
state, reward = right()
assert not done
assert reward == 0
assert state == C(i=0, j=2) and state != C(i=1, j=0)
state, reward = right()
assert not done
assert reward == 0
assert state == C(i=0, j=2) and state != C(i=1, j=0)
state, reward = right()
assert not done
assert reward == 0
assert state == C(i=0, j=2) and state != C(i=1, j=0)
state, reward = right()

assert state == C(i=1, j=1) and state != C(i=1, j=0)
state, reward = right()
assert not done
assert reward == 0
assert state == C(i=1, j=2) and state != C(i=1, j=0)
state, reward = right()
assert not done
assert reward == 0
assert state == C(i=1, j=2) and state != C(i=1, j=0)
state, reward = down()
assert not done
assert reward == 0
assert state == C(i=2, j=2) and state != C(i=1, j=0)
state, reward = right()
assert not done
assert reward == 0
assert state == C(i=2, j=2) and state != C(i=1, j=0)
state, reward = right()
assert not done
assert reward == 0
assert state == C(i=2, j=2) and state != C(i=1, j=0)
state, reward = right()
assert not done
assert reward == 0
assert state == C(i=2, j=2) and state != C(i=1, j=0)
state, reward = right()
```

