# OpenReview forum: "Large Language Models can Implement Policy Iteration"
_NeurIPS.cc/2023/Conference — NeurIPS 2023 poster_

### Official Review · Reviewer_EN6r · 2023-06-22

**Soundness:** 2 fair
**Presentation:** 2 fair
**Contribution:** 2 fair
**Rating:** 6
**Confidence:** 4

**Summary:**

This paper examines the possibility of taking advantage of the ability of in-context learning to use large language models to implement policy iteration so that large language models can be used to solve reinforcement learning problems without querying the gradients. Experiments have been conducted on 5 artificial environments showing that the proposed method ICPI can indeed achieve decent performance across all considered environments.

**Strengths:**

Given the success of in-context learning of large language models, applying in-context learning to policy iteration is a well-motivated idea. The proposed ICPI is sufficiently different from the original policy iteration and the other recent attempts to combine the success of large language models with reinforcement learning techniques. In the experiments, 5 artificial environments are proposed and ICPI achieves better results than baselines in all environments. Detailed ablations are also provided to justify the importance of each component of ICPI.


**Weaknesses:**

The applications of ICPI are limited. Right now it can only be applied to handcrafted small environments like chain and mini catch. It is unclear if large language models can do complex planning in more practical tasks such as Atari and mujoco tasks. Additionally, the current formulation of ICPI can only handle discrete state and action space. Still, it would be interesting to see if large language models can be used to handle realistic continuous-action robotics tasks. It also seems that a curated hint is required for more complicated tasks like mini invaders and point mass.

Comparisons to other more recent methods are missing. To justify the advantage of ICPI, it would be interesting to see how ICPI compares to popular deep RL baselines (either state-based or text-based) such as PPO [1], SAC [2], and CQL [3]/IQL [4] since some recent works [5, 6] have shown that they can be used for large language models too.

Some of the arguments are not supported. For example, in line 8-10, "All gradient techniques are inherently slow, sacrificing the 'few-shot' quality that makes in-context learning attractive to begin with" is not supported. ICPI also involves inferencing the entire trajectory to calculate the q function for each action, so it seems unclear whether it is faster than gradient-based techniques. A comparison of ICPI and gradient-based methods in terms of computation time is needed.

The writing can be improved. The related works and methodology sections can be more concise. There are some references typos and the paragraph from 160 - 164 is repeated.

[1] Proximal Policy Optimization Algorithms, https://arxiv.org/abs/1707.06347

[2] Soft Actor-Critic: Off-Policy Maximum Entropy Deep Reinforcement Learning with a Stochastic Actor, https://arxiv.org/abs/1801.01290

[3] Conservative Q-Learning for Offline Reinforcement Learning, https://arxiv.org/abs/2006.04779

[4] Offline Reinforcement Learning with Implicit Q-Learning, https://arxiv.org/abs/2110.06169

[5] Is Reinforcement Learning (Not) for Natural Language Processing: Benchmarks, Baselines, and Building Blocks for Natural Language Policy Optimization, https://arxiv.org/abs/2210.01241

[6] Offline RL for Natural Language Generation with Implicit Language Q Learning, https://arxiv.org/abs/2206.11871

**Questions:**

Please see weakness.

**Limitations:**

The limitation that current large language models lack the ability to plan in complex domains accurately is discussed.

---

> ### Author Rebuttal · Authors · 2023-08-04
>
> > The applications of ICPI are limited. Right now it can only be applied to handcrafted small environments like chain and mini catch. It is unclear if large language models can do complex planning in more practical tasks such as Atari and mujoco tasks. Additionally, the current formulation of ICPI can only handle discrete state and action space. Still, it would be interesting to see if large language models can be used to handle realistic continuous-action robotics tasks.
>
> This is an important question. Some recent work has demonstrated promise in applying large language models to complex tasks. For example, [Voyager: An Open-Ended Embodied Agent with Large Language Models](https://arxiv.org/abs/2305.16291) uses prompt engineering and hand-crafted state representations to elicit state-of-the-art policies from ChatGPT in the Minecraft domain, which is not continuous-action but still quite complex. A paper that was released very recently, [RT-2: Vision-Language-Action Models Transfer Web Knowledge to Robotic Control](https://arxiv.org/pdf/2307.15818.pdf), actually trains a video-language model to control a robot (with continuous actions) using text outputs.
>
> > Comparisons to other more recent methods are missing. To justify the advantage of ICPI, it would be interesting to see how ICPI compares to popular deep RL baselines (either state-based or text-based) such as PPO [1], SAC [2], and CQL [3]/IQL [4] since some recent works [5, 6] have shown that they can be used for large language models too.
>
> Thank you for directing our attention to these works. 5 and 6 are particularly interesting and will be integrated into our related works section. As for deep RL baselines, [this link](https://www.overleaf.com/read/ncjhbrxkvyrz) provides a demonstration of the performance of Proximal Policy Optimization (PPO) on the 5 domains in this paper. As the results indicate, most runs do not converge in less than 20,000 time-steps. ICPI converges in fewer than 250 timesteps on all 5 domains. We are working on generating results with SAC, but it is worth noting that SAC is not typically implemented for discrete action domains. As for CQL and IQL, offline reinforcement learning requires access to an offline dataset, so it is unclear how best to apply these algorithms to our domains. Please let us know how you would suggest approaching this.
>
> > Some of the arguments are not supported. For example, in line 8-10, "All gradient techniques are inherently slow, sacrificing the 'few-shot' quality that makes in-context learning attractive to begin with" is not supported. ICPI also involves inferencing the entire trajectory to calculate the q function for each action, so it seems unclear whether it is faster than gradient-based techniques. A comparison of ICPI and gradient-based methods in terms of computation time is needed.
>
> Again thank you for pointing this out. See our earlier comment and [results with PPO](https://www.overleaf.com/read/ncjhbrxkvyrz). Your point about computation time is well taken. Computing the ICPI results was quite slow because OpenAI places rate-limit restrictions on the Codex API. Given access to an unrestricted API, one would expect much faster results. Intuitively, one can expect faster performance from ICPI because it leverages knowledge already present in the LLM which the neural network must learn from scratch.
>
> > The writing can be improved. The related works and methodology sections can be more concise. There are some references typos and the paragraph from 160 - 164 is repeated.
>
> We appreciate the feedback and will correct these suggestions in the final version.

---

> > ### Comment · Reviewer_EN6r · 2023-08-10
> >
> > Thanks for your reply. I am now convinced by the comparison results with deep RL baselines and will update my score to 6. I do think that those are important baseline results so it would be great if the authors can include them in the final paper. Regarding the computation time, is it possible to calculate it based on open-sourced large language models such as OPT-30B as used in the paper (https://huggingface.co/facebook/opt-30b)?

---

> > > ### Author Response · Authors · 2023-08-10
> > > **Thank you**
> > >
> > > Thank you for updating your score. The suggestion to collect the computation time using OPT-30B makes sense. We will run these experiments over the weekend.

---

### Official Review · Reviewer_WHe7 · 2023-06-28

**Soundness:** 4 excellent
**Presentation:** 3 good
**Contribution:** 2 fair
**Rating:** 5
**Confidence:** 1

**Summary:**

This paper formalizes a framework where language model can use its in-context learning capability to predict actions and Q-values, and the training occurs by updating the prompt with the latest history. As a result, the LM can perform reinforcement learning without gradient. The paper explored a few language models and a few different toy tasks and found that Codex can learn successfully in this environment that it has no prior knowledge.

**Strengths:**

- The paper formalizes a framework where LLM can perform RL using its in-context learning capability. Such a framework might allow future works to connect LLM research to existing RL concepts to further improvement performance.
- The empirical results look promising.

**Weaknesses:**

I'm not an expert in RL, so I am not confident in my judgement.

- While the formalization seems nice, it is not particularly novel; the empirical results/datasets presented also seem toy and low-hanging fruits.

**Questions:**

N/A

**Limitations:**

The author properly listed their limitations.

---

> ### Author Rebuttal · Authors · 2023-08-04
>
> > While the formalization seems nice, it is not particularly novel; the empirical results/datasets presented also seem toy and low-hanging fruits.
>
> The Policy Iteration algorithm itself is, of course, many decades old. What is novel is the in-context implementation of policy iteration using LLMs. Because of limited access to LLMs because of the associated cost we could only illustrate our algorithmic ideas on simple domains. However, our algorithm is applicable to more complex domains without modification and furthermore we believe that as LLMs become more and more widely available and there is further improvement in their quality and generalization abilities our method will scale and generalize accordingly.

---

> > ### Comment · Reviewer_WHe7 · 2023-08-10
> > **Thanks for the clarification**
> >
> > I have read the response. Thanks for the clarification. As I'm not an expert in RL, please downweight my judgement on novelty.

---

> > > ### Author Response · Authors · 2023-08-10
> > > **Thank you**
> > >
> > > Thank you for the response. We understand the challenges related to reviewing a paper outside one's expertise. As this was one of your primary criticisms, would you consider increasing your score?

---

### Official Review · Reviewer_RKry · 2023-07-05

**Soundness:** 3 good
**Presentation:** 4 excellent
**Contribution:** 4 excellent
**Rating:** 7
**Confidence:** 3

**Summary:**

The authors show how LMs can solve simple MDPs by implementing policy iteration in-context. They do that by constructing prompts that simulate policy rollouts using partial trajectories from an experience buffer as few-shot examples. They empirically validate their approach on six toy MDPs that test for specific failure modes.

**Strengths:**

1. I find the result from the paper title impressive. It’s interesting from at least four different angles: (i) understanding the capabilities of LMs as general-purpose simulators, (ii) sketching the design space of hybrid systems combining RL with LMs, (iii) exploring prompting strategies for representing and rolling-out MDPs in token space, (iv) understanding the trade-offs between in-context learning and standard in-weights learning. The paper is a very original and solid contribution to all four.
2. The paper is well-written and easy to follow.
3. The experiments are pretty extensive, covering different MDPs (carefully designed to test specific failure modes), well-motivated ablations and different model families and sizes.

**Weaknesses:**

The weaknesses listed below are relatively minor and are greatly outweighed by strengths. The authors shouldn’t feel obliged to respond to them in detail.

1. The authors attribute the relatively good performance of ICPI its “ability to generalize from its context to unseen states and state/action pairs (unlike “Tabular Q” and “Matching Model”)” (lines 296-298). How is that different from function approximation in more traditional RL? It could've been good to also compare ICPI with some baseline using function approximation; both a variant with a small MLP and with a linear head on top of an LLM could be interesting.
2. The claim in the abstract that “algorithms that rely on [demonstrations] do not outperform the experts from which the demonstrations were derived” is too strong or too vague. It is possible in principle to generalise to rewards higher than those seen in the dataset and the DecisionTransformer paper demonstrated that in practice (in some cases; Fig. 4 in Chen et al.).
3. The Related work section is very comprehensive but also somewhat repetitive. If the authors’ are short of space, I think it could be shortened without much harm.

**Questions:**

1. What roles does the choice of LM sampling temperature=0.1 play in the setup? Was temperature=1 causing too much hallucination when generating rollouts? Does the theoretical convergence of policy iteration still hold when Q values are estimated from a  biased (due to temperature) distribution of rollouts?
2. What discount factor $\gamma$ is used in experiments? Is it 0.8, the same as for the optimal policy relative to which regret is computed?

**Limitations:**

Limitations and societal impacts are discussed. One class of risks that could be discussed is those associated with combining powerful world models with optimisation pressure for agentic behaviour (e.g., Ngo et al., 2023).

Ngo et al., 2023. The alignment problem from a deep learning perspective

---

> ### Author Rebuttal · Authors · 2023-08-04
>
> > The authors attribute the relatively good performance of ICPI its “ability to generalize from its context to unseen states and state/action pairs (unlike “Tabular Q” and “Matching Model”)” (lines 296-298). How is that different from function approximation in more traditional RL?
>
> Yes, this is a good question. Both our method and traditional RL function approximation methods take advantage of the generalization properties of deep neural networks. The primary difference is the data that these networks were trained on. For most RL methods, the policy and value networks are specialized to the environment or domain of interest, while our method relies on foundation models which are trained on a much larger and more general dataset. It is reasonable to expect that a traditional RL method will eventually outperform our method due to specialization, but it requires retraining on each new domain (while ICPI does not, since it can rely on the underlying foundation model for generalization).
>
> > Compare ICPI with some baseline using function approximation
>
> Thank you for pointing this out. This is clearly an omission in the paper. [This link](https://www.overleaf.com/read/ncjhbrxkvyrz) provides a demonstration of the performance of Proximal Policy Optimization (PPO) on the 5 domains in this paper. PPO is probably not the most sample-efficient RL algorithm, but as the results indicate, most runs do not converge in less than 20,000 time-steps. In contrast, ICPI converges in fewer than 250 timesteps. It is unlikely that another traditional deep RL method would significantly diminish this number. There are metalearning methods, like RL^2 or MAML, which use function approximation and which can few-shot-learn a domain, but these require a multitask setting, so it is unclear how to adapt them to our setting. Please let us know if there are other experiments we can run to address this concern.
>
> > The claim in the abstract that “algorithms that rely on [demonstrations] do not outperform the experts from which the demonstrations were derived” is too strong or too vague.
>
> This is valid. Perhaps we can state that “algorithms that rely exclusively on imitating demonstrations are inherently limited by the performance of the source expert.” In contrast, our method learns from new data generated by a continuously improving policy. We also draw attention to the fact that while Decision Transformer does outperform vanilla Behavior Cloning, it actually underperforms “Behavior Cloning 10%,” in which behavior cloning uses only the top 10% of trajectories. That can be thought of as similar to our “No ArgMax” which only includes trajectories that exceed a set performance threshold.
>
> > The Related work section is very comprehensive but also somewhat repetitive. If the authors’ are short of space, I think it could be shortened without much harm.
>
> Thank you for the suggestion. Several reviewers commented on the related works and we intend to rewrite it.
>
> > What roles does the choice of LM sampling temperature=0.1 play in the setup? Was temperature=1 causing too much hallucination when generating rollouts? Does the theoretical convergence of policy iteration still hold when Q values are estimated from a biased (due to temperature) distribution of rollouts?
>
> Even with temperature=0, the sampling of Q estimates is still stochastic because the prompts are sampled randomly and shuffled. Our analysis relies on the observation that, given a prompt containing transitions sampled from a mixture of policies and ordered randomly, the optimal prediction for a language model will be a mixture of these policies.
>
> > What discount factor is used in experiments? Is it 0.8, the same as for the optimal policy relative to which regret is computed?
>
> Yes to both questions.

---

> > ### Comment · Reviewer_RKry · 2023-08-15
> >
> > Thanks for a detailed response.
> >
> > > PPO is probably not the most sample-efficient RL algorithm, but as the results indicate, most runs do not converge in less than 20,000 time-steps. In contrast, ICPI converges in fewer than 250 timesteps
> >
> > Thanks, this is an insightful comparison. I wonder, however, how does this comparison look like if you measure the number of FLOPs required by both approaches, i.e. the FLOPs required by transformer forward passes during ICPI vs FLOPs required by forward passes, backwards passes and gradient updates involved in PPO training. Would the authors be able to do a back-of-the-envelope calculation to estimate that the order of magnitude of that difference?
> >
> > The bigger picture question is: how much less FLOP-efficient is ICPI compared with training a small NN-parametrised policy? I'm not expecting ICPI to be as FLOP-efficient (FLOP-efficiency is not the appeal of this method) but it seems interesting to know the size of the difference.

---

> > > ### Author Response · Authors · 2023-08-18
> > > **New link for accessing PPO (gradient-based) results**
> > >
> > > We have disabled our original results link to prevent de-anonymization.  Please use the following link to access the results: https://fromsmash.com/ppo-results

---

### Official Review · Reviewer_Yb53 · 2023-07-06

**Soundness:** 2 fair
**Presentation:** 3 good
**Contribution:** 2 fair
**Rating:** 5
**Confidence:** 3

**Summary:**

The paper demonstrates a way to implement policy iteration by means of in-context learning with a large language model. Authors suggest ICPI, a reinforcement learning algorithm that performs policy evaluation and policy improvement solely by encoding tokens and inferencing the model, without updating model parameters or adapters. They proceed by evaluating ICPI on a collection of simple RL problems, where each problem was manually converted to an LLM-digestible text representation. Authors show that certain ICPI configurations can achieve comparable or smaller regret than traditional RL algorithms, emphasizing the first few hundred steps. Finally, the paper analyzes the learned behavior produced by ICPI in the tested environments.

**Strengths:**

The main strength of the paper is that it provides a new way to think about policy iteration from an LLM perspective. While ICPI follows the general PI framework, the way it does so certainly "feels different" than traditional (e.g. tabular) PI. Applying similar ideas to other RL areas  may inspire interesting future research.

While the evaluation of ICPI is rather minimalistic, these experiments clearly showcase how the algorithm performs and explain the reported results in due detail.

Finally, the paper is generally well-written and easy to follow. The presentation quality is high, both for text and the accompanying visualizations.

**Weaknesses:**


### 1. Other methods have drawbacks... so what about them?

In the abstract / intro, authors review existing LLM applications in RL and broadly group them in two categories: those based on expert demosntrations, and those that use gradient-based updates. As a motivation, they state that each of these methods have drawbacks, and the drawback of gradient-based methods is that they are slow (e.g. L9).

To the best of my knowledge, there is nothing in the paper that evaluates if ICPI is "faster" than equivalently tuned gradient-based methods.
Furthermore, I have a hunch that ICPI scales worse than linearly with the number of demonstrations, i.e. 10000th episode is processed slower than the first one, due to the nature of Transformer attention.

My concern is that, since authors use speed as one of the motivating factors, this claim should be addressed somewhere in the paper. If ICPI is in fact slower than gradient-based methods, this should also be stated as a limitation. Please note that being slower does not discredit this work in any way -- but not exploring speed after using it as motivation does.

### 2. Comparison to other in-context reinforcement learning methods

There are at least several prior works that also fall in the category of "reinforcement learning in-context without updating model parameters".
One prior work that fits this definition is [1], where the LLM-based agent "reflects" on it's performance in natural language, which is shown to improve performance. Another, similar work [2] tasks the large language model with providing feedback on model performance in natural language, which also leads to improved behavior. It is likely that there are more prior & concurrent papers that are also relevant.

[1] https://arxiv.org/abs/2303.11366
[2] https://arxiv.org/abs/2303.17651

On a surface level, it seems that the above two papers take a less formalistic stance on policy improvement, relying on instruction-following LLMs to do most of their work. However, they have similar requirements and solve similar problems. Both papers use text-based RL environments of similar complexity to the ones used in this submission.

Since both these works were available for several months before submission, I would argue that this paper would benefit from comparing against them: both in terms of theoretical properties and in empirical regret on the same problem set.

**Questions:**



> L185 - since our primary results use codex language model, we use python code to represent these values

Minor question: was it an educated guess or does python outperform some alternative languages when used in this capacity?

> six domains and their associated prompt formats

Minor question: how robust is ICPI performance to the choice of prompts?


> L21: improve … as information accumulated

Possibly missing a verb, e.g. as information is accumulated / accumulates



> Fig 4 caption: Error bars are standard errors from four seeds

Assuming you use standard deviation, you may want to specify whether or not this is an adjusted estimate (i.e. whether or not you use Bessel’s correction). If you deliberately opt not to specify this, please feel free to ignore the question.


**Limitations:**

Authors properly address most of the main limitations of this work, as well as it's societal limitations.

One potentially missing limitation is the reproducibility of the main experiments. As it stands, ICPI relies on proprietary language models that can change without warning. To the best of my understanding, the current OpenAI policy allows them to alter the model behavior without warning users in advance, e.g. to patch a potential misuse. This could inadvertently affect the performance of ICPI, making it difficult to reproduce the main results in several years. While there is no guaranteed way to fix this issue, I would argue that it is worth acknowledging it.

---

> ### Author Rebuttal · Authors · 2023-08-04
>
> > To the best of my knowledge, there is nothing in the paper that evaluates if ICPI is "faster" than equivalently tuned gradient-based methods.
>
> Thank you for pointing out this omission. [This link](https://www.overleaf.com/read/ncjhbrxkvyrz) provides a demonstration of the performance of Proximal Policy Optimization (PPO) on the 5 domains in this paper. PPO is probably not the most sample-efficient RL algorithm, but as the results indicate, most runs do not converge in less than 20,000 time-steps. In contrast, ICPI converges in fewer than 250 timesteps on all 5 domains. We believe it is unlikely that another traditional deep RL method would diminish this number by several orders of magnitude. There are gradient-based metalearning methods, like RL^2 or MAML, which demonstrate learning on very small time scales, but these require a multitask domain, so it is unclear how to adapt them to our setting. Please let us know if there are other experiments we can run to address this concern.
>
> > Furthermore, I have a hunch that ICPI scales worse than linearly with the number of demonstrations, i.e. 10000th episode is processed slower than the first one, due to the nature of Transformer attention.
>
> In general, we do not believe that the algorithm will scale with the number of episodes. The time-complexity of the algorithm is linear in a) the number of actions b) the expected length of an episode c) the size (especially the context window and layer count) of the foundation model. It is true that better policies may generate longer episodes for some environments, but episode length does not usually scale linearly with the optimality of the policy.
>
> > My concern is that, since authors use speed as one of the motivating factors, this claim should be addressed somewhere in the paper. If ICPI is in fact slower than gradient-based methods, this should also be stated as a limitation.
>
> Again thank you for pointing this out. See our earlier comment and [results with PPO](https://www.overleaf.com/read/ncjhbrxkvyrz). Please let us know if other experimental results would better address this concern.
>
> > Minor question: was it an educated guess or does python outperform some alternative languages when used in this capacity?
>
> We did not have the opportunity to compare against other languages due to the high cost of running these experiments. Our expectation would be that the LLM's performance would roughly scale with the prevalence of a given language in its dataset. We note that the Codex paper focuses its evaluations almost exclusively on python.
>
> > Minor question: how robust is ICPI performance to the choice of prompts?
>
> In our ablation experiments, we do demonstrate several variations on the prompt selection mechanism proposed in the paper. Most demonstrate reasonable performance while underperforming the proposed method. Perhaps you are specifically interested in the wording of the prompt. Our experience during experimentation was that the language model is indifferent to details of punctuation or syntax, but sensitive to the semantics of the information present in the prompt, as demonstrated especially by our “no hints” ablation.
>
> > Possibly missing a verb, e.g. as information is accumulated / accumulates
>
> Thank you for the correction. We will fix this in the final version.
>
> > Assuming you use standard deviation, you may want to specify whether or not this is an adjusted estimate (i.e. whether or not you use Bessel’s correction). If you deliberately opt not to specify this, please feel free to ignore the question.
>
>  Our plots use the Pandas [“rolling.sem()” method](https://pandas.pydata.org/docs/reference/api/pandas.core.window.rolling.Rolling.sem.html) with a rolling window of 30.
>
> > One potentially missing limitation is the reproducibility of the main experiments. As it stands, ICPI relies on proprietary language models that can change without warning. To the best of my understanding, the current OpenAI policy allows them to alter the model behavior without warning users in advance, e.g. to patch a potential misuse.
>
> This is a good point and one that we will include in the limitations section. It is worth noting that, since the submission of this paper, all newer models (the GPT-4 family) have only improved on the capabilities of the earlier codex model that we were using.

---

> > ### Comment · Area_Chair_8Cde · 2023-08-14
> >
> > Thanks everyone!
> >
> > To Reviewer Yb53: Can you please take a look at the author response and let us know if the author response addresses your concerns about Weaknesses 1 and 2?
> >
> > To the authors: any additional thoughts about the relationship of this work to some of the less-formal reflection-based methods the reviewer mentioned above?

---

> > > ### Author Response · Authors · 2023-08-15
> > > **Reflection-based methods.**
> > >
> > > Yes thank you for drawing our attention to this point. Both [1] and [2] are very valuable contributions and works of great interest. [1] and [2] and our paper all present ideas for "iterative self-improvement" of LLMs. That said, the details and settings are quite different.
> > > - [2] is not in an embodied setting and uses self-improvement to improve on standard types of language tasks, while we implement a form of policy improvement (a form or RL) in embodied tasks.
> > > - Our work draws a more general connection between foundation models and the theory of policy iteration, which has been extensively studied and applied in the literature on reinforcement learning.
> > > - [1] and [2] rely on the ability to clearly verbally describe a policy. Our approach, which acts directly on state-action pairs, is more general and may have greater utility in settings such as robotics where such a verbal description is difficult.
> > > - Finally, we should note that our work has been publicly available on Arxiv since 7 Oct 2022, whereas [1] was posted on 20 Mar 2023 and [2] was posted on 30 Mar 2023. To the best of our knowledge, neither of these works have been published in an official venue.

---

> > > ### Comment · Reviewer_Yb53 · 2023-08-18
> > > **Response by the reviewer**
> > >
> > > Authors mostly addressed my first and main concern by providing additional comparisons. As such, I raise my score to reflect this change, assuming that authors will consider modifying their paper to include these additional experiments (regardless of the final decision). I also thank authors for answering my questions.
> > >
> > > There are, however, two remaining issues that prevent me from further rising the score. I summarize them below and leave it to Area Chain 8Cde to determine their significance:
> > >
> > > 1. **In-context reinforcement learning.** In my original review, I raised a concern (see "2. Comparison to other in-context reinforcement learning methods"). To the best of my understanding, authors chose not to address this concern.
> > >
> > > 2. **Possible violation of discussion rules.** Authors provide some of their results by means of an overleaf link. The text draft itself is reasonably anonymized. However, some of the overleaf features may be in violation of anonymity policy. For instance, simultaneous document editors would be able to see each other in the top-right corner of the overleaf page. Aside from this, authors essentially submit a 7-page PDF response (via overleaf) which may or may not violate the discussion rules about a single-page PDF submission.
> > >
> > > To be clear,
> > > - I do not suspect any deliberate malicious intent by the authors from submitting this link
> > > - I took reasonable precautions to preserve anonymity (e.g. I viewed the project as a throwaway account)
> > >
> > > Still, I recommend AC to clarify this matter with the program committee.

---

> > > > ### Author Response · Authors · 2023-08-18
> > > > **Comparison to other in-context reinforcement learning methods**
> > > >
> > > > The author's previous post was intended to address this concern. In particular we addressed [1] https://arxiv.org/abs/2303.11366 and [2] https://arxiv.org/abs/2303.17651:
> > > > - [2] is not in an embodied setting and uses self-improvement to improve on standard types of language tasks, while we implement a form of policy improvement (a form or RL) in embodied tasks.
> > > > - Our work draws a more general connection between foundation models and the theory of policy iteration, which has been extensively studied and applied in the literature on reinforcement learning.
> > > > - [1] and [2] rely on the ability to clearly verbally describe a policy. Our approach, which acts directly on state-action pairs, is more general and may have greater utility in settings such as robotics where such a verbal description is difficult.
> > > > - Finally, we should note that our work has been publicly available on Arxiv since 7 Oct 2022, whereas [1] was posted on 20 Mar 2023 and [2] was posted on 30 Mar 2023. To the best of our knowledge, neither of these works have been published in an official venue.

---

> > > > > ### Comment · Area_Chair_8Cde · 2023-08-18
> > > > >
> > > > > PCs have confirmed that the new link is OK and there will be no impact on decisions. We appreciate everyone's quick response here. Authors, please start with an anonymous link in the future.

---

> > ### Comment · Area_Chair_8Cde · 2023-08-18
> > **Re: Overleaf**
> >
> > Hi everyone,
> >
> > Reviewer Yb53: thanks for bringing up the overleaf issue. I'm discussing this with PCs right now. In the meantime:
> >
> > Authors: please, as soon as possible, replace all of the Overleaf links in your review with anonymous direct links to the PDF in question. Because an Overleaf account is required to view the file, and accounts are tied to email addresses, the current response runs a serious risk of violating the anonymity policy. I'd appreciate it if you could reply here as soon as you've edited the review.

---

> > > ### Author Response · Authors · 2023-08-18
> > > **Correcting overleaf links.**
> > >
> > > We apologize for the oversight. We are working on replacing the links now.

---

> > > > ### Author Response · Authors · 2023-08-18
> > > > **New link for accessing PPO results.**
> > > >
> > > > The overleaf links will no longer work. Instead we are using a file sharing service called "smash." Please use the following link to access the results: https://fromsmash.com/ppo-results

---

> > > > > ### Comment · Area_Chair_8Cde · 2023-08-18
> > > > >
> > > > > Thank you for taking care of this so quickly!

---

### Official Review · Reviewer_6gCR · 2023-07-08

**Soundness:** 3 good
**Presentation:** 2 fair
**Contribution:** 2 fair
**Rating:** 5
**Confidence:** 4

**Summary:**

The problem of using large-pretrained foundation models (LLMs) like codex as backbones for learning RL policies without gradient updates or expert demonstrations is addressed by the authors. They propose an algorithm called In-Context Policy Iteration (ICPI) to perform model-based RL policy iterations using a pretrained LLM to predict state/rewards as well as the next-action.

To make the environment prompt-ready, the task is translated into a text-based description. The authors focus on code-pre-trained backbones and consider translating to Python code in their experiments. They demonstrate that ICPI can learn to minimize regret over time in simple tabular environments such as maze and point-mass.

**Strengths:**

Some strengths of the work are outlined below:
1. The proposed algorithm utilizes In-Context learning to implement policy-iteration with LLMs. This parameter-free approach has the potential to enhance the task-specific behavior of an LLM agent by designing a prompt-ready text-based interface for any MDP.
2. The authors present a clear explanation of the algorithm for ICPI, including the necessary design choices. They also provide empirical evidence of policy improvement over time in six MDPs that are both simple and tabular.

**Weaknesses:**

1. Based on the results, it appears that the algorithm works efficiently when combined with code-pre-training. However, relying solely on text-based MDP descriptions may not yield consistent outcomes. It is uncertain whether this is due to the limitations in reasoning and in-context learning capabilities of current LLMs, or the difficulty in creating effective intermediate representations of the environment.
2. To perform Q-value computation, the algorithm needs to unroll action sequences for all actions. Unfortunately, this greatly restricts the algorithm's usefulness to finite action-spaces.

**Questions:**

1. The success of policy-iteration using replay-buffer enhanced prompting is likely influenced by the information content of the demonstration set $\mathcal{D}$. It's worth noting that these trajectories may not be on-policy and are sourced from previous policy versions. It's important to investigate how the algorithm responds to the quality of the replay-buffer, including factors like the number of trajectories and whether failed examples are necessary.

**Limitations:**

The authors have acknowledged certain limitations of their work and discussed them thoroughly. One such limitation is that the ICPI method requires exhaustive unrolling of the action sequence for all actions, making it suitable only for domains with finite action space. Furthermore, it appears that only code-pre-trained models are capable of reliably performing ICPI, indicating that translating the MDP to an appropriate code format is crucial. Additionally, the method often relies on additional hints to ground the reward function/terminal states, hence a more comprehensive discussion on potential ways to mitigate this would be beneficial.

---

> ### Author Rebuttal · Authors · 2023-08-04
>
> > To perform Q-value computation, the algorithm needs to unroll action sequences for all actions. Unfortunately, this greatly restricts the algorithm's usefulness to finite action-spaces.
>
> You are correct that we only demonstrated the method using discrete action domains. However we suggest two considerations. First, many existing state-of-the-art methods, like Dreamer-v2 or Muzero work primarily (or exclusively) on discrete-action domains, some of which (like Go or Minecraft) can be quite complex. Second, a variety of publications have demonstrated that continuous action domains can often be addressed through discretization (e.g. [Trajectory Transformer](https://arxiv.org/abs/2106.02039) and [RT-1: Robotics Transformer for Real-World Control at Scale](https://arxiv.org/abs/2212.06817)).
>
> > The success of policy-iteration using replay-buffer enhanced prompting is likely influenced by the information content of the demonstration set. It's worth noting that these trajectories may not be on-policy and are sourced from previous policy versions. It's important to investigate how the algorithm responds to the quality of the replay-buffer, including factors like the number of trajectories and whether failed examples are necessary.
>
> It is true that the trajectories may not be strictly on-policy. First of all, we can provide no strict guarantees on the output of a pre-trained LLM. Second, some elements of the policy prompt are drawn from slightly older trajectories, which may be generated by an older policy. To mitigate this, we only draw the policy prompt from the c=8 most _recent trajectories_. In our “c=16” baseline, we investigate the result of extending the recency cutoff and allowing for trajectories generated by older policies. As Figure 3 indicates, the algorithm is still able to learn, though this modification hurts performance. As for failed examples, they are necessary for the world model component of the algorithm, but not the policy component, which in general benefits from the best demonstrations available.
>
> > Based on the results, it appears that the algorithm works efficiently when combined with code-pre-training. However, relying solely on text-based MDP descriptions may not yield consistent outcomes. It is uncertain whether this is due to the limitations in reasoning and in-context learning capabilities of current LLMs, or the difficulty in creating effective intermediate representations of the environment.
>
> Perhaps you  can offer some clarification here. It sounds like you are asking about whether any remaining sub-optimality is attributable to the LLM itself or to the text format that we chose. Is this correct?

---

> > ### Comment · Area_Chair_8Cde · 2023-08-14
> >
> > Thanks everyone! Reviewer 6gCR, could you please clarify the last point in the author response?
> >
> > As an alternative for the authors, it would be helpful for me (and probably other reviewers) to get an answer to both interpretations of the question.

---

> > > ### Author Response · Authors · 2023-08-18
> > > **Response to both interpretations of author's question**
> > >
> > > > Based on the results, it appears that the algorithm works efficiently when combined with code-pre-training. However, relying solely on text-based MDP descriptions may not yield consistent outcomes. It is uncertain whether this is due to the limitations in reasoning and in-context learning capabilities of current LLMs, or the difficulty in creating effective intermediate representations of the environment.
> > >
> > > Coming up with effective intermediate representations of an environment can be a challenge, especially in text, although several recent publications have scaled this approach to some surprisingly difficult problems (consider [1] https://arxiv.org/abs/2305.16291 and [2] https://arxiv.org/pdf/2207.05608.pdf). That said, one might imagine settings to which other formats, such as images, are better suited. The main requirement of our approach is that the model is capable of modeling sequences in the given format.
> > >
> > > Most of the suboptimality that we observe in our approach is likely due to the inability of the current generation of LLMs to infer the correct pattern from the given text trajectories. We observed that in the paper models older than Codex were entirely incapable of performing this kind of inference.

---

### Decision · Program_Chairs · 2023-09-21

**Decision:**

Accept (poster)

**Comment:**

This paper studies the use of neural language models for policy learning, finding that LLMs can learn to implement policy iteration in-context rather than relying on fine-tuning. All reviewers rated the paper positively, with at least one particularly enthusiastic.

The authors have promised some changes to the discussion of related work, as well as new experimental results, which should be incorporated into the final version of the paper. If there's room, it would also be nice to include some discussion of reviewer RKry's question about tradeoffs between sample efficiency (rollouts) and computational efficiency (FLOPs).